# Ischemic Postconditioning Regulates New Cell Death Mechanisms in Stroke: Disulfidptosis

**DOI:** 10.3390/biom14111390

**Published:** 2024-10-31

**Authors:** Shanpeng Liu, Qike Wu, Can Xu, Liping Wang, Jialing Wang, Cuiying Liu, Heng Zhao

**Affiliations:** 1Laboratory of Brain Disorders, Beijing Institute of Brain Disorders, Ministry of Science and Technology, Joint Innovation Center for Brain Disorders, Capital Medical University, Beijing 100069, China; shanpengliu@mail.ccmu.edu.cn (S.L.); qikewu@mail.ccmu.edu.cn (Q.W.); wlp19980317@163.com (L.W.); 201202101@njnu.edu.cn (J.W.); 2Yunnan Key Laboratory of Southern Medicine Utilization, College of Chinese Materia Medica, Yunnan University of Chinese Medicine, Kunming 650500, China; 15208703065@163.com; 3School of Nursing, Capital Medical University, Beijing 100069, China; liucy@mail.ccmu.edu.cn

**Keywords:** disulfidptosis, stroke, IPostC, *PRDX1*, bioinformatics

## Abstract

Background and Objective: Stroke poses a critical health issue without effective neuroprotection. We explore ischemic postconditioning’s (IPostC) potential to mitigate stroke-induced brain injury, focusing on its interaction with disulfidptosis, a novel cell death pathway marked by protein disulfide accumulation. We aim to clarify IPostC’s protective mechanisms against stroke through gene sequencing and experimental analysis in mice. Methods: Through our initial investigation, we identified 27 disulfidptosis-related genes (DRGs) and uncovered their interactions. Additionally, differential gene analysis revealed 11 potential candidate genes that are linked to disulfidptosis, stroke, and IPostC. Our comprehensive study employed various analytical approaches, including machine learning, functional enrichment analysis, immune analysis, drug sensitivity analysis, and qPCR experiments, to gain insights into the molecular mechanisms underlying these processes. Results: Our study identified and expanded the list of disulfidptosis-related genes (DRGs) critical to stroke, revealing key genes and their interactions. Through bioinformatics analyses, including PCA, UMAP, and differential gene expression, we were able to differentiate the effects of stroke from those of postconditioning, identifying *Peroxiredoxin 1* (*PRDX1*) as a key gene of interest. GSEA highlighted *PRDX1*’s involvement in protective pathways against ischemic damage, while its correlations with various proteins suggest a broad impact on stroke pathology. Constructing a ceRNA network and analyzing drug sensitivities, we explored *PRDX1*’s regulatory mechanisms, proposing novel therapeutic avenues. Additionally, our immune infiltration analysis linked *PRDX1* to key immune cells, underscoring its dual role in stroke progression and recovery. *PRDX1* is identified as a key target in ischemic stroke based on colocalization analysis, which revealed that *PRDX1* and ischemic stroke share the causal variant rs17522918. The causal relationship between *PRDX1*-related methylation sites (cg02631906 and cg08483560) and the risk of ischemic stroke further validates *PRDX1* as a crucial target. Conclusions: These results suggest that the DRGs are interconnected with various cell death pathways and immune processes, potentially contributing to IPostC regulating cell death mechanisms in stroke.

## 1. Introduction

Stroke continues to pose a significant global health challenge, resulting in high mortality rates and disability [1,2,3]. It results in extensive brain cell death, accompanied by immune response and inflammation, which are therapeutic targets in stroke treatment [4,5,6]. We have studied the protective effect of ischemic postconditioning (IPostC) in stroke research since 2006 [7,8]. One possible mechanism by which IPostC exerts its effects is through the inhibition of the generation of free radicals, neuroinflammation, and apoptosis, ultimately resulting in a reduction in infarct size [7]. Nevertheless, the protective mechanisms of IPostC remain largely unknown. Thus, it is essential to elucidate the mechanisms of IPostC.

Disulfidptosis is a novel cell death mechanism, an emerging form of programmed cell death. It is marked by an atypical accumulation of disulfides, which are chemical bonds formed between sulfur atoms within proteins [9,10,11]. This leads to irregular disulfide connections in actin cytoskeletal proteins, culminating in cell contraction and death [12,13]. Disulfidptosis occurs due to the elevated expression of *solute carrier family 7 member 11* (*SLC7A11*) expression under glucose deprivation, initiating cell death through reduction–oxidation reactions and the formation of disulfur bonds [14,15].

Our research delved into the association between disulfidptosis and stroke. One group discovered that, in ischemic stroke patients, there was a significant upregulation in the expression of several disulfidptosis-related genes, such as *IQGAP1*, *TLN1*, *CAPZB*, *INF2*, and *SLC7A11*, compared to controls [16]. Our group has also performed disulfidptosis analysis in stroke patients. Through gene analysis of patient blood samples, we found that these DRGs are linked with various immune cells and cell death pathways [17]. This suggests that DRGs may play a contributory role in the pathology of stroke. Despite its potential significance, the role of disulfidptosis in the protective effects of IPostC has not yet been fully elucidated, and there have been no studies using IPostC animal models to explore DRGs in stroke.

Leveraging RNA sequencing data from our mouse stroke models, we probed the protective impact of IPostC on DRG expression in cerebral tissues. Our analysis focused on validating DRG expression, understanding their role in disulfidptosis and immune modulation, and assessing their diagnostic and therapeutic prospects. This inquiry lays the groundwork for future explorations into disulfidptosis mechanisms and their therapeutic potential in enhancing stroke patient care through IPostC.

## 2. Materials and Methods

### 2.1. Animal Model

Adult male *C57BL*/*6 mice* 22–28 g were purchased from the Department of Zoology, Capital Medical University (Beijing, China). Mice were fed with standard chow and fed in individual cages containing free drinking water. The living environment of the experimental animals was maintained in the temperature range of 22–24 °C and relative humidity in the range of 60–70%. Circadian rhythms and sound and light stimuli were avoided. All animal experiments were approved by the Institutional Animal Care and Use Committee (IACUC: AEEI-2022-203) of Capital University of Medical Sciences and conformed to the principles outlined in the National Institutes of Health Guide for the Care and Use of Laboratory Animals. In this study, all possible measures were taken to minimize animal suffering and animal numbers.

### 2.2. Transient Focal Cerebral Ischemia and Ischemic Postconditioning

Surgery was performed based on our previous studies [7,18,19] (Figure 1). We induced focal ischemia by occluding the left middle cerebral artery (MCA) for 45 min. To induce IPostC, we utilized brief, repetitive MCA occlusion through suture withdrawal and re-insertion into the internal carotid artery [7,20,21]. Temporal reperfusion was achieved by withdrawing the suture approximately 2 to 3 mm and subsequently re-inserting it. IPostC was initiated 2 min after reperfusion and involved repeated 3 cycles of 15 s of occlusion followed by 30 s of reperfusion.

### 2.3. Sampling and Sequencing

After 24 h of reperfusion, brain tissues were dissected from the same regions of the ipsilateral side of the uninfarcted, peri-ischemic, and ischemic core regions from the sham-operated, MCAo, and IPostC groups, respectively. The peri-infarct region is the ischemic tissue preserved by post-treatment, whereas the ischemic core is the infarcted area in the ischemic brain treated with IPostC. There were five biological replicates for each of the three groups of mice, and 15 fresh brain tissue samples were frozen in liquid nitrogen at −80 °C. All sequencing technologies were provided by the company (Novogene, Beijing, China). In addition, we used samples (three biological replicates) for further experimental validation.

### 2.4. Analyzing Workflow

Our analyses (shown in Figure 1) included differential gene expressions (DEGs), machine learning, immune infiltration, gene function enrichment, drug sensitivity analyses, real-time fluorescence quantitative PCR, and ceRNA network studies to elucidate the molecular mechanisms underlying stroke and postconditioning protection.

### 2.5. Identification of DRGs

We reviewed the recent literature to identify eighteen genes linked to disulfidptosis [15]. We then used the GeneMANIA database to expand the genes to 40 and analyzed the potential interactions and functions of these disulfide-related genes (DRGs) [22] (http://www.genemania.org, accessed on 12 November 2023). GeneMANIA helped prioritize and validate these genes by revealing functional associations and shared pathways.

Furthermore, we employed the STRING database to examine co-expressed DRGs, both upregulated and downregulated, to determine their pathways and functions [23] (https://string-db.org/, accessed on 13 November 2023). Using Cytoscape, networks were created to visualize the strengths of gene associations and interactions based on a confidence threshold of >0.4 [24]. These bioinformatics approaches helped us to reliably characterize DRGs and understand their roles in disulfidptosis.

### 2.6. Functional Enrichment Analysis of DRGs and Peroxiredoxin 1 (PRDX1)

In our study, we conducted a functional enrichment analysis leveraging both KEGG and GO annotations to explore the pathways and molecular functions associated with DRGs. To enrich the gene set function analysis, we utilized the KEGG REST API to access the most current KEGG Pathway gene annotations. These annotations served as a background for mapping the genes within our collection. The enrichment analysis was carried out using the clusterProfiler R package (version 3.14.3), which facilitated the identification of significant gene set enrichments. We established thresholds for the gene sets at a minimum of 5 and a maximum of 5000 genes, with *p*-values less than 0.05 deemed to indicate statistical significance [25]. Similarly, GO annotations were analyzed using the org.hs.egg.db R package (version 3.1.0), applying comparable methodologies to ascertain significant molecular functions and biological processes.

We conducted a *PRDX1* Single Gene Set Enrichment Analysis (GSEA) utilizing the GSEA software (version 3.0) to stratify our samples into two categories based on *PRDX1* expression levels: high (≥50%) and low (<50%). This classification allowed us to explore the underlying biological pathways and mechanisms differentially activated between these groups. To achieve this, we employed the Molecular Signatures Database (MSigDB) as a comprehensive resource for pathway and mechanism annotations, correlating gene expression patterns with phenotypic variations [26] (http://www.gsea-msigdb.org/gsea/downloads.jsp, accessed on 23 November 2023). Our analysis focused on identifying significantly overrepresented gene sets, utilizing normalized enrichment scores (NESs) to quantify the degree of overrepresentation in each category. We rigorously controlled the false discovery rate (FDR) to minimize the likelihood of type I errors, applying the Benjamini and Hochberg correction method. Pathways and mechanisms were deemed statistically significant if they met the criteria of an FDR < 0.25 combined with an absolute NES value greater than 1 [27]. This approach ensured that the identified pathways and mechanisms were both relevant and reliable, highlighting the critical roles they play in the context of *PRDX1* expression levels.

### 2.7. Differential Gene Expression Analysis

We applied the limma method, utilizing the R package limma (version 3.40.6), to identify genes linked to stroke and IPostC based on differential expression. With a cutoff of |logFC| > 1.1 and *p*-value < 0.05, we pinpointed significant differentially expressed genes (DEGs). This approach enabled us to isolate DEGs with notable fold changes and statistical relevance, illuminating the molecular dynamics related to stroke and IPostC and enhancing our comprehension of their underlying mechanisms and potential therapeutic targets.

### 2.8. Combining Random Forest and LASSO Regression for Gene Selection and Identification of Key Genes

We utilized the Random Forest algorithm, recognized for its precision and reliability, through the “randomForest” R package [28]. To improve prediction accuracy and feature selection, LASSO regression was implemented using the “glmnet” R package, known for its ability to select crucial variables by applying regularization [29]. By optimizing the model with a Lambda value of 0.09, we successfully identified the top five predictive genes.

### 2.9. Extraction of RNA and qRT-PCR

We used MolPure Cell/Tissue Total RNA Kit to extract RNA from ischemic penumbra brain tissue. The cDNA was then synthesized using the NovoScript Plus All-in-one 1st Strand cDNA Synthesis SuperMix on the gene amplification apparatus (TC-E-480). Quantitative RT-PCR was performed by LightCycler 480II (Roche, Shanghai, China). The mRNA levels of target genes were normalized to endogenous β-Actin. They were expressed and quantified by 2^−ΔΔct^. The primer sequences (F 5′-GCCGCTCTGTGGATGAGATTA-3′, R 5′-AGCTGGACACACTTCACCAT-3′) for *PRDX1* quantitation were taken from the literature [30].

### 2.10. CeRNA Network Construction

The miRTarBase database (https://mirtarbase.cuhk.edu.cn/, accessed on 23 November 2023) was used to predict the target miRNAs of the biomarkers [31]. Then, the ENCORI database (https://rnasysu.com/, accessed on 23 November 2023) screening criteria: CLIP data ≥ 1, CLIP region *p*-value < 0.05) to predict circRNAs targeting miRNAs. circRNA-miRNA-mRNA networks were constructed using Cytoscape (version 3.10.1) [32].

### 2.11. PRDX1 Drug Sensitivity Analysis

We utilized the RNAactDrug database (http://biobigdata.hrbmu.edu.cn/RNAactDrug/index.jsp, accessed on 26 November 2023) a comprehensive platform, to investigate the relationship between drug sensitivity and various RNA molecules (including mRNA, lncRNAs, and miRNAs). This database provides insights across four molecular dimensions: expression, copy number variation, mutation, and methylation, integrating data from three principal pharmacogenomic databases: GDSC, CellMiner, and CCLE. For our analysis, we set an FDR threshold of 0.01 to identify compounds linked to *PRDX1*. Utilizing Spearman statistics, we then employed Cytoscape (version 3.10.1) to visualize the association between drug sensitivity and *PRDX1* expression. Additionally, we sourced medication datasets from the CellMiner database (https://discover.nci.nih.gov/cellminer/home.do, accessed on 23 November 2023). to further examine this relationship. Using Cytoscape (version 3.10.1), we visualized the correlation between *PRDX1* sensitivity and various drugs, focusing on those with an FDR of <0.05 to underscore significant associations.

### 2.12. Analysis of Immune Cell Infiltration

We assessed the immune landscape and cell-specific characteristics within the Sham, MCAO, and IPostC groups using the XCELL algorithm [33,34]. This involved exploring the expression of critical immune checkpoint genes such as *PDCD1*, *HAVCR2*, *CD86*, *BTN2A2*, *CD274*, *CD28*, *LGALS9*, *TNFRSF9*, *TNFSF14*, *TNFSF9*, *VTCN1*, *H13*, and *PVR* to predict their impact on therapeutic efficacy. Furthermore, we employed Spearman’s analysis to establish connections between immune cell infiltration, immune checkpoints, and key genes. The relationships were visualized using a “correlogram,” aiming to provide a detailed overview of the immune cell composition and gene interactions. This comprehensive examination seeks to shed light on the role of the immune system in the pathology of ischemic stroke and IPostC [35].

### 2.13. Data Preparation and Screening Criteria for Colocalization Analysis

This analysis was conducted to determine whether the same genetic variants influence both *PRDX1* expression and the risk of ischemic stroke, providing insights into potential shared genetic mechanisms underlying the disease. Colocalization analysis using IEU Open GWAS data (https://gwas.mrcieu.ac.uk/, accessed on 24 June 2024) (Appendix A) and coloc (version 5.2.3) package evaluated associations between *PRDX1* and ischemic stroke. The Bayesian approach evaluated five hypotheses: H0 (no SNP associated with any trait in the region), H1 (a causal SNP associated with the first trait), H2 (a causal SNP associated with the second trait), H3 (a causal SNP associated with both traits), and H4 (presence of two different SNPs, each corresponding to one trait). We specified the a priori probability (p1, p2) of the SNP being associated with only a single trait as 1 × 10^−4^. Strong colocalization evidence (PH4 > 0.8, p12 > 1 × 10^−4^) indicated shared causal variation, offering insights into relationships between genetic signals at specific loci and ischemic stroke [36]. This analysis leveraged the comprehensive GWAS summary data available through the IEU Open GWAS project to provide a mechanistic understanding of the genetic basis underlying complex phenotypes like ischemic stroke.

### 2.14. Mendelian Randomization Analysis on DNA Methylation of PRDX1 and Stroke Risk

We performed this analysis to determine if DNA methylation at the *PRDX1* gene has a causal impact on stroke risk by leveraging genetic variants associated with these methylation sites. We looked for methylation sites of *PRDX1* through the EWAS Open Platform (https://ngdc.cncb.ac.cn/ewas/datahub/, accessed on 30 June 2024) (Appendix A) [37]. The study obtained DNA methylation quantitative trait loci (mQTLs) regulating DNA methylation sites of *PRDX1* from the DNA Methylation Consortium (GoDMC), an international collaboration of human epidemiological studies with over 30,000 participants providing genetic and DNA methylation data. mQTLs data were obtained from the publicly available resource at http://fileserve.mrcieu.ac.uk/mqtl/assoc_meta_all.csv.gz (accessed on 30 June 2024) [38]. For stroke outcome data, the study used the R10 version of the comprehensive Finnish database (https://www.finngen.fi/, accessed on 30 June 2024) with 399,220 participants of European descent [39]. Ethical approval was obtained from local ethics committees, and no new data collection was involved. SNPs associated with mQTLs data were selected with a threshold of *p* < 1 × 10^−5^, and LD checks were performed (r2 = 0.1, kb = 100). The f-statistic was calculated for each SNP to evaluate Instrumental Variable Selection (IVs) strength and mitigate weak instrument bias, excluding those with F values below 10 as instrumental variables [40]. The study verified that the selected exposures satisfied three crucial assumptions: (1) exposure–outcome correlation, (2) no confounding variables linking IVs and exposure–outcome relationship, and (3) exclusive impact of IVs on the outcome through exposure without other mechanisms. Any IVs violating these assumptions were excluded from the analysis. The study employed TwoSampleMR (version 0.6.1) and MR-PRESSO (version 1.0) packages in R (4.2.1) for MR analysis, including IVW, MR-Egger, weighted median, weighted mode, and MR-PRESSO methods [41,42]. Results were expressed as odds ratio (OR) with 95% confidence interval (CI), considering *p* < 0.05 as statistically significant. MR-Egger and MR-PRESSO were used to assess horizontal pleiotropy, and Cochran’s Q test evaluated heterogeneity among selected SNPs (*p* < 0.05 indicated heterogeneity) [43]. MR-PRESSO was able to identify and address outliers beyond horizontal pleiotropy [44]. Standard errors and Confidence Internal (CI) were calculated using delta methods.

### 2.15. Statistical Analysis and Visualization of Results: R Package Usage and Significance Evaluation

R version 4.2.1 was used for statistical analyses and graph visualization in this study. The ggplot2 software package (version 3.3.6) was employed to create the visualization graphs. Correlation network graphs were generated using igraph (version 1.4.1) and ggraph (version 2.1.0). Additionally, several other bioinformatic analyses were conducted. For ROC analysis, the qROC package (version 1.18.0) was utilized. The Wilcoxon rank sum test was employed to assess differences between groups, with statistical significance set at *p* < 0.05 (* *p* < 0.05; ** *p* < 0.01; *** *p* < 0.001).

## 3. Result

### 3.1. Exploration and Definition of DRGs

Based on a thorough examination of recent studies [15], we have carefully chosen 18 specific genes that play crucial roles in disulfide processes, including *SLC7A11*. GeneMANIA analysis expanded the initial set of 18 DRGs to a total of 40 genes by considering their mutual relationships (Figure 2A). This analysis also revealed similar functional characteristics among the DRGs. Additionally, STRING analysis identified 112 significant interactions among 27 of these gene-expressed proteins (Figure 2B). We summarize the possible relationship between 27 genes and disulfidoptosis (Appendix A).

To further explore the functional implications of these genes, GO and KEGG analyses were performed. The results indicated their involvement in processes such as actin organization and pathways, including ferroptosis (Figure 2C,D). The establishment of these 27 DRGs set the stage for the subsequent exploration of the relationship between disulfidoptosis and stroke postconditioning.

### 3.2. DEGs Analysis to Identify Key Genes in Stroke and IPostC

Dimensionality reduction via PCA and UMAP on 15 samples revealed distinct clustering and significant inter-group differences (Appendix A). Differential gene expression analysis identified 5606 upregulated and 986 downregulated genes in MCAo vs. SHAM (Figure 3A) and 561 upregulated and 1454 downregulated genes in IPostC vs. MCAo (Figure 3C,D), with heatmaps and volcano plots illustrating key differences. Wayne plots revealed 1549 genes affected by both MCAo and IPostC (Figure 3E) and pinpointed 11 genes at the intersection of MCAo, IPostC, and DRGs for further study (Figure 3F).

### 3.3. PRDX1 Is Identified as the Cornerstone of IPostC’s Protective Mechanisms Uncovered by Expression Analysis and Machine Learning of DRGs

Analysis of the DRGs further reveals that between the MCAo and SHAM groups, 11 DRG genes show significant differential expression, while between the IPostC and MCAo groups, 5 DRG genes differ significantly (Figure 4A, all *p* < 0.05). Machine learning algorithms, including LASSO and Random Forest, have confirmed and prioritized these genes’ significance, identifying *PRDX1* as the key gene aligning most closely with our model of interest (Figure 4B–D).

### 3.4. Diagnostic Value of Candidate Genes and mRNA Expression Validation of PRDX1

We evaluated key genes as potential diagnostic markers for stroke and IPostC by plotting ROC curves, revealing that five genes distinctly associated with MCAo and IPostC changes could differentiate between conditions, with all ROC curves (Figure 5A,B) showing a predictive AUC > 0.7. Focusing on *PRDX1* due to its superior model fit, RT-qPCR confirmed its increased expression post-MCAo and suppression after IPostC treatment (Figure 5C), highlighting *PRDX1*’s diagnostic potential and regulatory role in treatment response.

### 3.5. GSEA Analysis of PRDX1 in Stroke and IPostC

GSEA analysis of the *PRDX1* gene across four databases highlighted 20 key pathways critical to IPostC, revealing significant enrichment in pathways like ECM (extracellular matrix), Receptor Interaction and Calcium Signaling Pathway in the KEGG database (Figure 6A), Regulation of Macrophage-Derived Foam Cell Differentiation and T-Cell Differentiation in the GO-BP database (Figure 6B), components of the plasma membrane in the GO-CC database (Figure 6C), and functions like Hexokinase Activity in the GO-MF database (Figure 6D), all indicating *PRDX1*’s downregulation and its protective role in IPostC (|NES| > 1.3, NP < 0.05).

### 3.6. Prediction and Expression Correlation Analysis of PRDX1-Related Proteins

To gain insight into the role of *PRDX1* in stroke and IPostC, we used the STRING website to predict 10 genes significantly associated with *PRDX1* protein expression and show their interaction networks (Figure 7A). These genes are *ABI1*, *PRDX1*, *PRDX2*, *PRDX5*, *SRXN1*, *TXN2*, *TXN1*, *TXNRD1*, *GPX1*, *CDK5*, and *CDK5R1*. The correlation coexpression network diagram showed that *CDK5R1* was negatively correlated with *PRDX1*, *SRXN1*, *GPX1*, and *TXNRD1*. *GPX1* and *SRXN1* showed a strong positive correlation with *PRDX1* coexpression (Figure 7B). These results suggest that *PRDX1* and several genes may participate in the pathological mechanism and protective effect of IPostC, which lays a foundation for further research on the mechanism of *PRDX1*.

### 3.7. Investigating PRDX1-Related Protein Networks and Expression Correlations

To elucidate the involvement of *PRDX1* in stroke and ischemic postconditioning (IPostC), we utilized the STRING database to identify and visualize the interaction network of 10 genes closely associated with *PRDX1* protein expression. These genes include *ABI1*, *PRDX1*, *PRDX2*, *PRDX5*, *SRXN1*, *TXN2*, *TXN1*, *TXNRD1*, *GPX1*, *CDK5*, and *CDK5R1*. Our coexpression network analysis revealed that *CDK5R1* exhibited a negative correlation with *PRDX1*, *SRXN1*, *GPX1*, and *TXNRD1*, while *GPX1* and *SRXN1* demonstrated a strong positive correlation with *PRDX1* (Figure 7B). These findings imply that *PRDX1*, alongside several identified genes, may play crucial roles in the pathological processes and protective mechanisms of IPostC, providing a foundation for further mechanistic studies on *PRDX1*’s function.

Additionally, we explored the drug sensitivities associated with the *PRDX1* gene, integrating the top 30 drugs into a network analysis. This network included 31 nodes (representing 1 gene and 30 drugs) and established 30 connections. Notably, drugs such as 6-methoxy-benzothiazol-2-yl-carbamic acid ethyl ester and 1-Amino-3-(methylthio) propyl (methyl) phosphinic acid exhibited significant negative correlations with *PRDX1* expression (Figure 8B). These findings not only shed light on potential pathways regulated by *PRDX1* but also suggest promising avenues for future research into its role in enhancing stroke treatment and recovery strategies.

### 3.8. Analysis of Immune-Related Aspects of PRDX1

Immune infiltration is closely related to stroke development, and a large number of recent results have been reported in the literature [45,46,47,48,49]. Immune infiltration analysis underscores its significance in stroke progression and postconditioning recovery, with the xCell algorithm highlighting differences in immune cell composition and microenvironment scores across MCAo, SHAM, and IPostC groups (Figure 9A). Specifically, MCAo samples showed increased macrophages but reduced keratinocytes and neurons, while SHAM and IPostC samples exhibited the opposite trend (Figure 9B). *PRDX1*’s correlation with these immune cells suggests its involvement in stroke’s immune dynamics, showing strong positive and negative correlations with macrophages and keratinocytes/neurons, respectively (Figure 9C–E). Consistent results exist for other key genes, including *ACTB*, *APBB1IP*, *CYFIP1*, and *PGD* (Appendix A). Additionally, *PRDX1*’s positive correlations with immune checkpoint genes *CD86* and *PVR* (*PVR* Cell Adhesion Molecule, also known as poliovirus cellular receptor) underscore its key immunomodulatory role in stroke and postconditioning (Figure 9F–H), hinting at IPostC’s potential protective effects on peripheral immunity in stroke.

### 3.9. Colocalization between PRDX1 and Ischemic Stroke

Single nucleotide polymorphism (SNP) refers to the existence of two or more base variations at a single nucleotide in the genome, widely distributed and can locate genetic loci associated with complex phenotypes, providing an important tool for studying the genetic basis of diseases [50].

Colocalization analysis can help identify shared causal SNPs associated with both *PRDX1* and ischemic stroke [51]. We found *PRDX1* with colocalization results with nine fracture data points, and we provided specific details of the associated data (Appendix A). Colocalization analysis was performed, and the results are summarized in Appendix A (PP.H4 = 1.00). *PRDX1* and nine ischemic strokes were found to be rs17522918 in all GWAS colocalization analyses (Figure 10A–I). These findings have important implications for understanding disease pathogenesis and could inform the development of targeted interventions to prevent or manage ischemic stroke.

### 3.10. Causal Effects of PRDX1-Related Methylation Sites on Ischemic Stroke

The study found that two methylation sites (cg02631906 and cg08483560) associated with the *PRDX1* gene were significantly linked to the risk of ischemic stroke and showed a causal relationship. As shown in Appendix A, two methylation sites associated with the *PRDX1* gene (cg02631906 and cg08483560) remained significant under the MR Hypothesis and were identified as cause-and-effect associated with ischemic stroke according to the IVW method. There was a causal relationship between cg02631906 and ischemic stroke (OR: 1.07, 95% CI: 1.01–1.14, *p* = 0.03). There was a causal relationship between cg08483560 and ischemic stroke (OR: 1.04, 95% CI: 1.00–1.07, *p* = 0.01). Our findings also suggest an increased role of two methylation sites in the risk of ischemic stroke. The scatter plot (Figure 11A,E) shows an assessment of the causal effects of the relationship between methylation sites and the risk of ischemic stroke. Forest maps of the causal effects of single cg02631906 and CG08483560-associated single nucleotide polymorphisms on ischemic stroke were observed in the MR_SingleSNP test (Figure 11B,F). The reservation-one test showed that no SNPS with large effect sizes had a bias in the estimates (Figure 11C,G). Global tests of IVW and MR-Egger indicated that heterogeneity is unlikely (Figure 11D,H).

## 4. Discussion

In this study, we identified a novel set of 27 disulfidptosis-related genes (DRGs) associated with stroke and IPostC, advancing our understanding of the molecular mechanisms underlying stroke pathology and neuroprotection. Among these, *PRDX1* emerged as a pivotal gene, playing a crucial role in the oxidative stress response during stroke. Our analysis demonstrated significant upregulation of *PRDX1* in the ischemic stroke model, with modulation by IPostC, underscoring its potential as a therapeutic target. Furthermore, GSEA revealed that *PRDX1* is involved in key protective pathways, such as extracellular matrix receptor interaction and calcium signaling, which are essential in mediating the effects of IPostC. Additionally, we identified a shared causal variant (rs17522918) between *PRDX1* and ischemic stroke, establishing a genetic link that supports *PRDX1*’s role in stroke pathology. Our methylation analysis further identified two *PRDX1*-related methylation sites (cg02631906 and cg08483560) that show a causal relationship with stroke risk, highlighting the epigenetic regulation of *PRDX1*. Moreover, immune infiltration analysis suggested that *PRDX1* influences both stroke progression and recovery, potentially impacting immune-related therapeutic strategies. Collectively, these findings provide novel insights into the role of disulfidptosis and *PRDX1* in stroke, offering new avenues for therapeutic intervention and underscoring the diagnostic and therapeutic potential of *PRDX1* in ischemic stroke.

Through our analysis using GO and KEGG, we discovered the involvement of the identified DRGs in processes such as actin cytoskeleton organization, actin filament-based processes, and ferroptosis. These findings provide scientific validity and support the relevance of the identified DRGs. In our study, we employed comprehensive bioinformatics analysis, including DEG analysis, functional enrichment analysis, immune analysis, and drug sensitivity analysis, to investigate the relationship between disulfidptosis and stroke, particularly in the context of IPostC. Our focus was on brain immune cells associated with cell death mechanisms. Firstly, we identified a set of DRGs associated with disulfidptosis, suggesting their potential involvement in stroke and IPostC. Additionally, we identified a group of hub genes, namely *PRDX1*, *ACTB* (*actin beta*), *APBB1IP* (*amyloid beta precursor protein binding family B member 1 interacting protein*), *CYFIP1* (*cytoplasmic FMR1 interacting protein 1*), and *PGD* (*hosphogluconate dehydrogenase*), which exhibited high expression in stroke and significant downregulation of gene expression under IPostC. To further validate our findings, we employed machine learning techniques to define the most critical gene, *PRDX1*. Additionally, we conducted qRT-PCR experiments to confirm the differential expression results in stroke and IPostC. These findings provide valuable insights into the relationship between disulfidptosis and stroke, particularly in the context of IPostC. The integration of multiple bioinformatics analyses, along with experimental validation, strengthens the scientific rigor of our study.

Recent research highlights the increased expression of *Peroxiredoxin 1* (*PRDX1*) in stroke patients, suggesting its role in oxidative stress response and potential as a stroke biomarker [46,52,53]. However, *PRDX1* plays a dual role in stroke, involving both protective and damaging effects. Protective roles have been demonstrated through enhanced antioxidant gene expression, reducing ischemic brain injury and improving neural cell resilience to oxidative stress [54,55,56,57]. Studies highlight *PRDX1*’s antioxidative defense mechanism in mitigating stroke damage, emphasizing its potential as a therapeutic target. Conversely, research also reveals *PRDX1*’s damaging role in ischemic–reperfusion injury, promoting neuroinflammatory damage through the *TLR4/NF-κB* pathway and exacerbating stroke outcomes [58]. These findings underscore *PRDX1*’s complex involvement in stroke, offering insights for novel treatment strategies targeting its regulation to alleviate post-stroke neuroinflammation. Although the role of *PRDX1* in ischemic stroke has been mentioned in other studies, our research provides fresh insights by identifying key genes and networks related to stroke, especially within the context of IpostC.

The mechanisms behind *PRDX1*’s dual role in stroke-induced brain injury remain elusive. This dichotomy might stem from its varied activity across different time points, in distinct cell types, and across various models. Initially, *PRDX1*’s antioxidative actions serve as a crucial defense, safeguarding brain tissue in the early phases of ischemic injury by reducing oxidative stress [59,60]. Yet, its later involvement in triggering pro-inflammatory pathways, notably the *TLR4*/*NF-κB* axis, during extended ischemic conditions potentially aggravates the injury [58]. This dual functionality of *PRDX1* reflects the adaptive nature of cellular responses to stress—what begins as a protective mechanism may evolve into a source of damage when prolonged or misdirected. Understanding the temporal and contextual nuances of *PRDX1*’s actions is fundamental for developing stroke therapies that capitalize on its protective effects while minimizing inflammatory harm.

We recognize that *PRDX1* expression did not increase in the IPostC group compared to the MCAo group, likely due to the specific IPostC model used. Despite this, *PRDX1* was chosen for further analysis due to its critical role in the oxidative stress response, a key factor in stroke pathology. Although *PRDX1* levels were reduced in the IPostC group, they remained elevated compared to the SHAM group, indicating ongoing residual damage. The GSEA analysis was conducted to explore the pathways associated with *PRDX1* under different conditions, providing insights into its continued relevance in the context of IPostC and its potential as a diagnostic and therapeutic target.

The discovery that IPostC inhibits *PRDX1* is significant as it sheds light on a potential therapeutic mechanism for reducing stroke-induced brain injury. This finding is reasonable considering that the timing and context of *PRDX1* activity are critical; by modulating *PRDX1* during optimal windows, IPostC could offer a targeted approach to stroke treatment [53,60]. This balance between antioxidative protection and avoidance of excessive inflammation could be key in improving recovery outcomes after a stroke, making our findings a promising avenue for future therapeutic strategies.

In addition to *PRDX1*, study results suggest that *ACTB*, *CYFIP1*, *APBB1IP*, and *PGD* also possess diagnostic and target values. *ACTB*, a key cytoskeletal component, is crucial in stroke pathology as its disruption impairs neuronal structure and function, affecting stroke outcomes and recovery [61,62]. *CYFIP1* plays a crucial role in stroke recovery by enhancing synaptic plasticity and dendritic remodeling through its regulation of *CAMK2A* expression, with disruptions in *CYFIP1* impairing these neuroprotective effects [63]. *APBB1IP* (*RIAM*) is crucial for the proper activation and function of integrins in neutrophils, with its absence leading to impaired phagocytosis and its dysregulation potentially impacting stroke-induced neuronal injury and repair [64]. *PGD*, an enzyme in the pentose phosphate pathway, is essential for managing oxidative stress [65]. Altered *PGD* expression can influence stroke outcomes by affecting the cell’s oxidative stress response. Collectively, these genes are critical for understanding the mechanisms by which IPostC regulates stroke and for identifying potential therapeutic targets.

SNPs provide a tool for locating genetic loci associated with complex phenotypes, enabling the study of the genetic basis of diseases. Colocalization analysis identified *PRDX1* and ischemic stroke as sharing the causal variant rs17522918, which has important implications for understanding the disease’s pathogenesis and developing targeted interventions. The study also highlights the causal effects of *PRDX1*-related methylation sites (cg02631906 and cg08483560) on ischemic stroke, emphasizing their increased role in the risk of the condition. These findings contribute to our understanding of the genetic basis of ischemic stroke and may guide preventive and management strategies.

One of the limitations of our study is the use of young adult mice as the animal model for investigating ischemic stroke and the effects of IPostC. Stroke predominantly affects older individuals who often present with various comorbidities, including obesity, hypertension, and diabetes, which can significantly influence the pathophysiology and outcomes of the disease [66]. The young adult mice used in our experiments do not fully replicate these age-related factors, which may limit the translational relevance of our findings to the clinical setting. Specifically, the absence of age-related changes and comorbid conditions in our animal model may lead to differences in immune response, cellular aging, and recovery mechanisms compared to older, comorbid human patients. Consequently, while our study provides valuable insights into the molecular and cellular mechanisms of IPostC, further research using aged animal models that better reflect the clinical population is necessary to fully understand the potential therapeutic benefits of IPostC in stroke patients.

## 5. Conclusions

In conclusion, our study elucidates stroke’s molecular mechanisms, identifying key genes and their roles in disease progression, with a focus on *PRDX1*’s protective actions against stroke via disulfidptosis and IPostC mechanisms. These insights enhance our understanding of stroke mitigation strategies. Though further research is required for validation, our findings mark a significant step forward in stroke research and potential therapeutic developments.

## Figures and Tables

**Figure 1 biomolecules-14-01390-f001:**
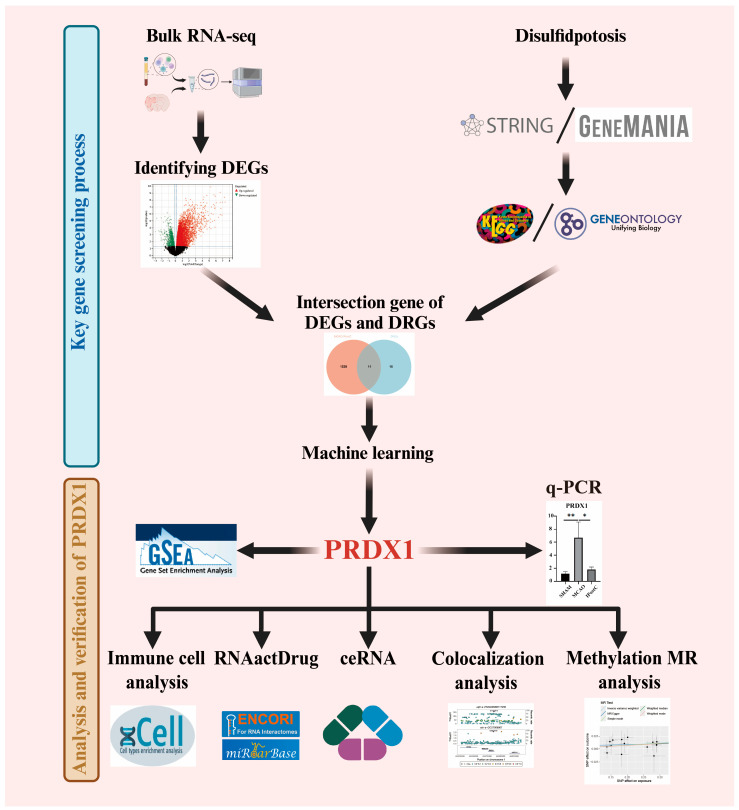
Workflow and data processing steps. The figure illustrates the sequential analysis pipeline used in the study. It is divided into two parts: the screening of key genes and the analysis and verification of key genes. It also presents all the methods and software used in the analysis, including data sources, differential gene expression analysis, functional enrichment analysis, key gene screening, real-time fluorescent quantitative PCR validation, drug sensitivity analysis, and immune cell analysis.

**Figure 2 biomolecules-14-01390-f002:**
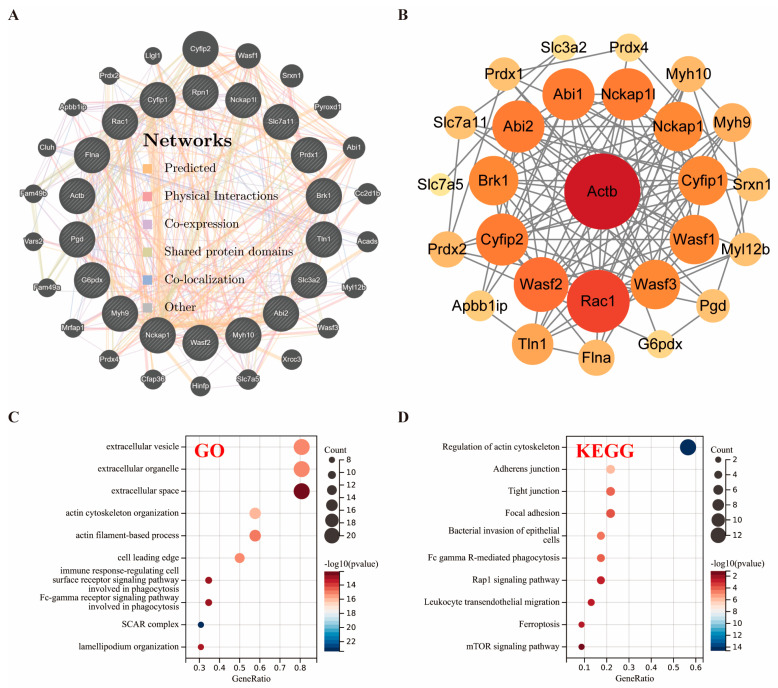
Reasonable DRG extensions and definitions. (**A**) Identify 40 DRGs with similar functions by the GeneMANIA website. The colors of the lines represent the types of gene interactions, including shared protein domains, predicted, physical interactions, coexpression, colocalization, and other. (**B**) Interaction of 27 DRGs binding proteins by the STRING database. The network consists of 112 edges and 27 nodes, the size of which is determined by the importance (degree) of the protein. Bubble diagram of the KEGG and GO (BP, CC, and MF) terms associated with DRGs. (**C**) KEGG; (**D**) GO (all *p* < 0.05).

**Figure 3 biomolecules-14-01390-f003:**
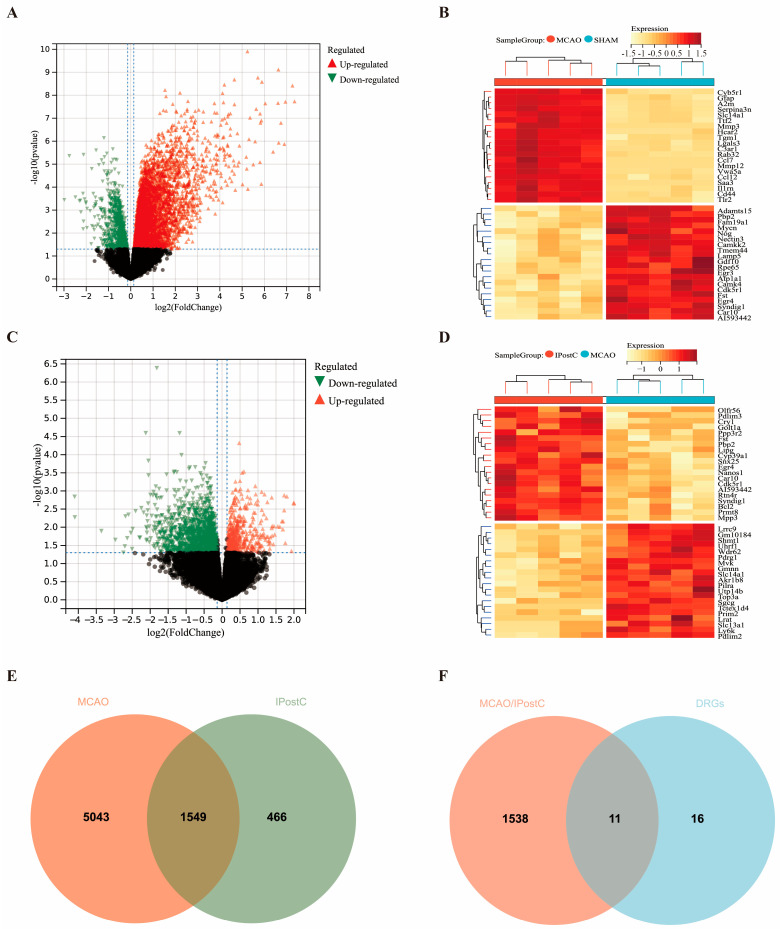
Identification and filtering of DEGs and DRGs in MCAo and IPostC models. This figure presents the comparison of gene expression between the Sham and MCAo groups through (**A**) a volcano plot and (**B**) a heatmap, as well as between the MCAo group and the IPostC group using (**C**) a volcano plot and (**D**) a heatmap, applying a threshold of |log2 (fold change)| > 1.1 and *p*-value < 0.05. The heatmaps illustrate the expression levels of genes, with darker hues representing a stronger positive expression correlation and lighter hues indicating a stronger negative correlation. Specifically, the heatmaps highlight the top 20 upregulated and downregulated genes. Volcano plots visually differentiate gene expression between MCAo and Sham samples and between MCAo and IPostC samples, respectively. Panel (**E**) displays a Venn diagram illustrating the intersection of DEGs between the MCAo and IPostC groups, identifying a total of 1549 overlapping DEGs. Panel (**F**) features a Wayne diagram showing the overlap of DEGs between the MCAo/IPostC dataset and DRGs, revealing 11 intersecting genes.

**Figure 4 biomolecules-14-01390-f004:**
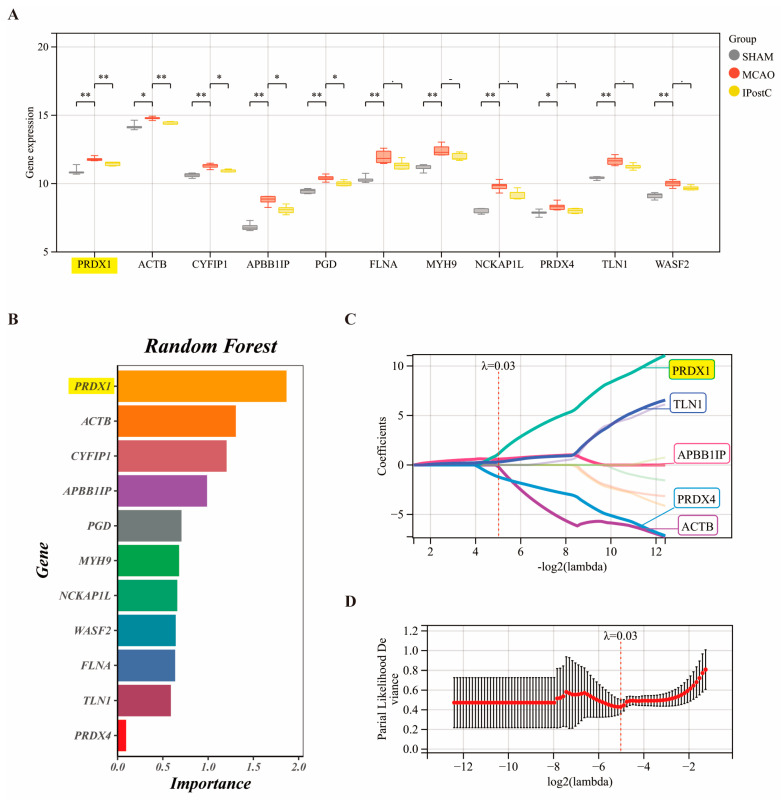
Expression patterns of DRGs across treatment modalities and identification of key genes via machine learning. (**A**) Highlights the variance in expression levels of the 11 identified DRGs across three experimental conditions: Sham, MCAo, and IPostC. (**B**) Utilizes Random Forest analysis, and (**C**,**D**) employs LASSO regression to pinpoint the most critical central gene: *PRDX1* (NA *p* > 0.05; * *p* < 0.05; ** *p* < 0.01).

**Figure 5 biomolecules-14-01390-f005:**
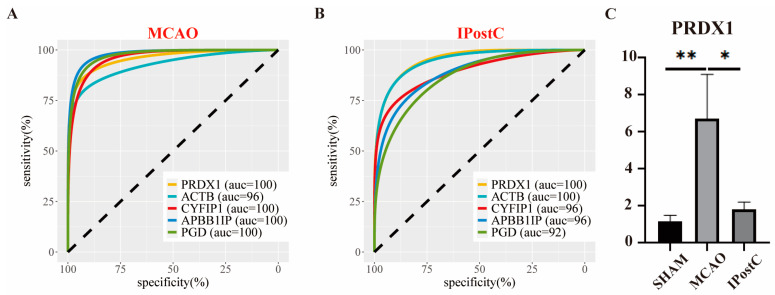
Evaluating the diagnostic potential of candidate genes and RT-qPCR confirmation of *PRDX1* expression. (**A**) The Receiver Operating Characteristic (ROC) curve demonstrates the diagnostic efficacy of selected genes in identifying stroke. (**B**) ROC curve analysis to assess the diagnostic capability of key genes following IPostC intervention. (**C**) Validation of *PRDX1* mRNA expression levels through RT-qPCR (NA *p* > 0.05; * *p* < 0.05; ** *p* < 0.01).

**Figure 6 biomolecules-14-01390-f006:**
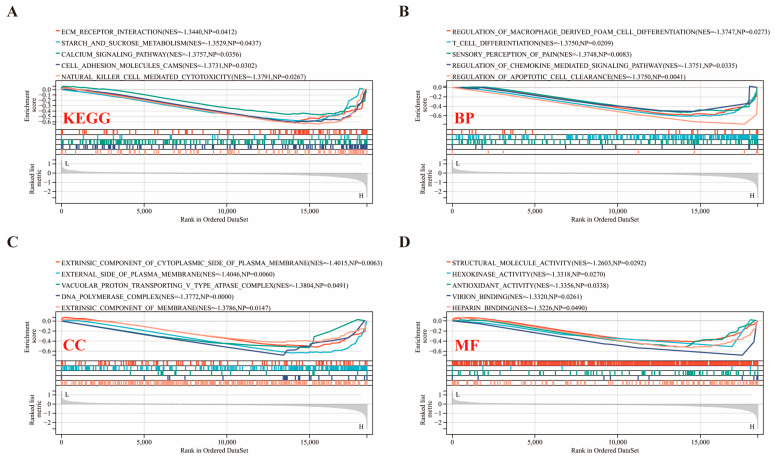
Unraveling the link between *PRDX1* and biological pathways in IPostC’s protective mechanism via GSEA. Employing GSEA analysis across various databases revealed (**A**) insights from the KEGG database, (**B**) analysis based on GO: biological processes (BPs), (**C**) findings from GO: cellular components (CCs), and (**D**) results pertaining to GO: molecular functions (MFs). NP, nominal *p*-value; NES: normalized enrichment score.

**Figure 7 biomolecules-14-01390-f007:**
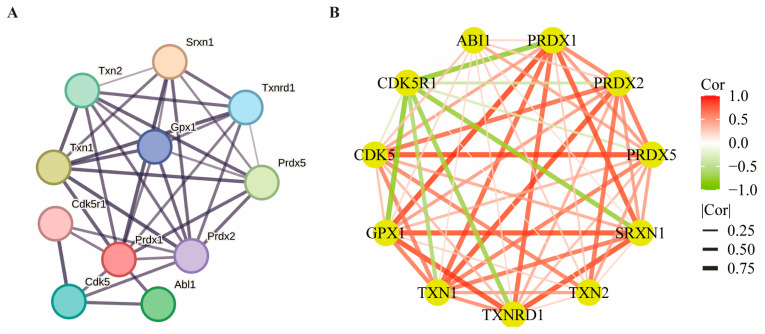
Prediction and expression correlation analysis of *PRDX1*-related proteins. (**A**) *PRDX1*-associated binding protein interactions were identified by the STRING database. The network consists of 35 edges and 11 nodes (PPI enrichment *p*-value = 9.01 × 10^−9^). (**B**) Based on transcriptome sequencing data, a correlation network map for predicting protein gene expression was constructed, with nodes representing genes and edges representing correlations. Red represents positive correlation, and green represents negative correlation, and the thicker the edge, the stronger the correlation.

**Figure 8 biomolecules-14-01390-f008:**
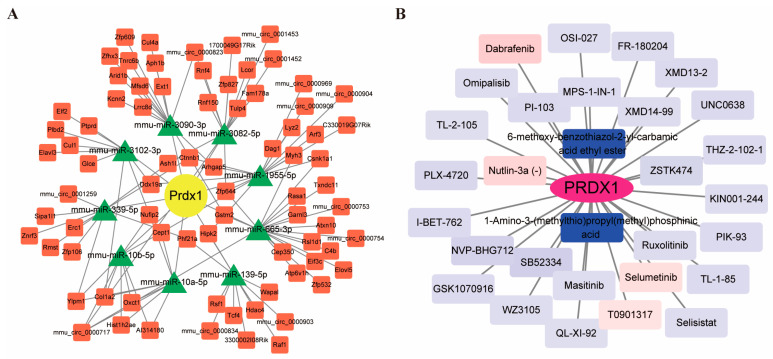
*PRDX1*-associated ceRNA network construction and drug sensitivity network construction. (**A**) Single-gene ceRNA network prediction plot, where red rectangles represent cyclic RNA, green triangles represent miRNA, and yellow circles represent gene. The gray lines represent the interactions of the components in the network diagram. (**B**) *PRDX1* drug sensitivity network diagram, where the rectangle is the compound, the purple oval represents the mRNA, the blue of the rectangle represents the negative correlation, the red represents the positive correlation, and the darker the color, the stronger the correlation.

**Figure 9 biomolecules-14-01390-f009:**
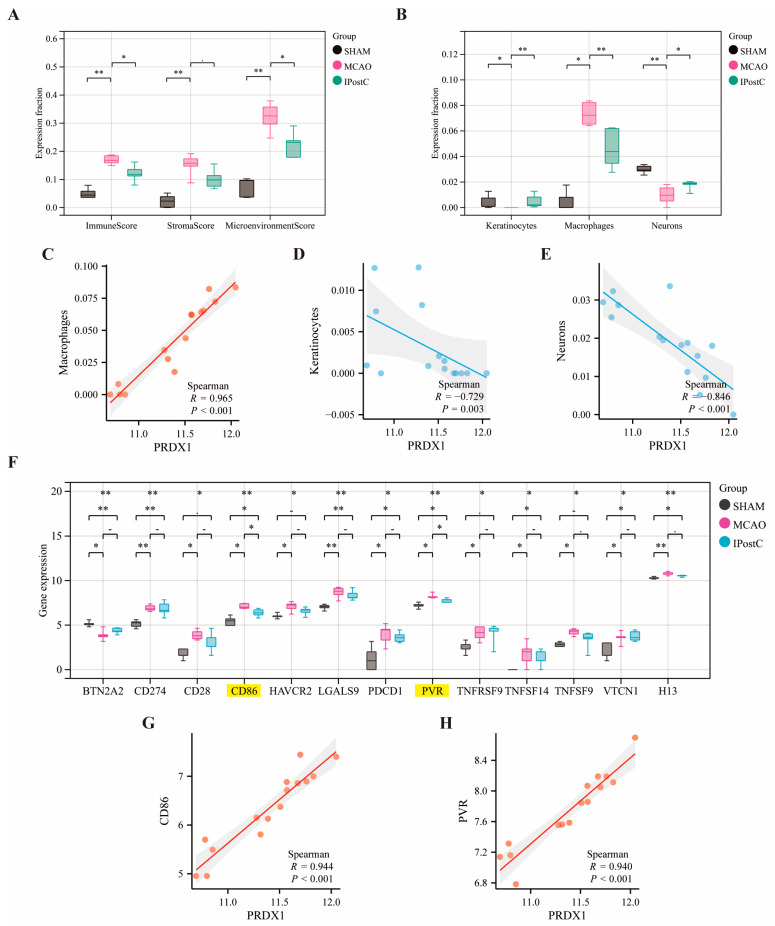
Immune landscape and checkpoint gene correlation with *PRDX1* expression. (**A**) Box plots depict the distribution of immune score, stromal score, and microenvironment score across SHAM, MCAo, and IPostC groups. (**B**) Box plots compare the relative abundance of keratinocytes, macrophages, and neurons. Colors represent different groups: SHAM (black), MCAo (pink), and IPostC (green). Scatter plots illustrate the correlation between *PRDX1* expression and (**C**) macrophages, (**D**) keratinocytes, and (**E**) neurons, with red indicating a positive correlation and blue a negative correlation. Spearman’s correlation coefficients (R) and *p*-values indicate statistical significance. (**F**) Box plots show the expression levels of mouse immune checkpoint genes across the SHAM, MCAo, and IPostC groups, with gray for SHAM, rose red for MCAo, and blue for IPostC. (**G**,**H**) CD86 and PVR demonstrate a strong positive correlation with *PRDX1* expression. Significance levels are indicated by asterisks (NA *p* > 0.05; * *p* < 0.05; ** *p* < 0.01).

**Figure 10 biomolecules-14-01390-f010:**
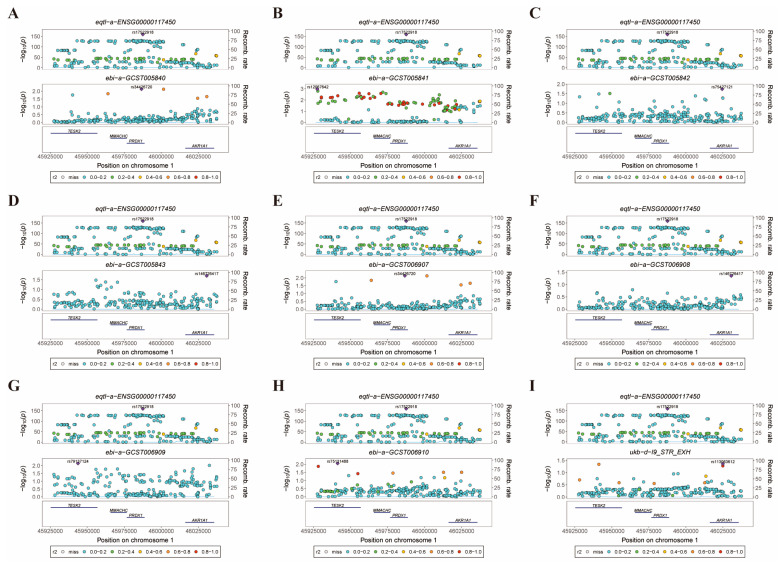
Locus comparing plots for the shared causal variant for the associations of *PRDX1* with ischemic stroke. The colocalization analysis revealed a shared causal variant for the associations of *PRDX1* with ischemic stroke in the gene region (Chr1: 45976723-45988562), which is located within ±50 kb from rs17522918. Within this region, rs17522918 was identified as the lead variant in the eQTL (eqtl-a-ENSG00000117450) of *PRDX1* and showed a strong correlation with the lead variant identified in the GWAS of ischemic stroke. Labels for GWAS data: (**A**) ebi-a-GCST005840, (**B**) ebi-a-GCST005841, (**C**) ebi-a-GCST005842, (**D**) ebi-a-GCST005843, (**E**) ebi-a-GCST006907, (**F**) ebi-a-GCST006908, (**G**) ebi-a-GCST006909, (**H**) ebi-a-GCST006910, and (**I**) finn-b-I9_STR_EXH. The r2 value represents the linkage disequilibrium (LD) between the variants and the top SNPs, indicating their genetic association.

**Figure 11 biomolecules-14-01390-f011:**
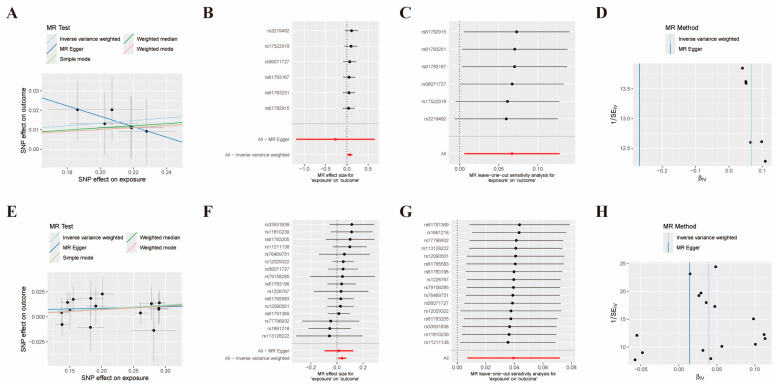
MR Analysis of methylation sites of *PRDX1* and risk of ischemic stroke. The scatter plot shows the size (β) of the SNP effect on the outcome (*Y*-axis) and exposure (*X*-axis), 95% confidence interval. Each dot represents an SNP that is used as a genetic tool. The slope represents the estimate for each of the five different MR Tests. The SNP effect of each methylation site is expressed as an SD per 1 unit high, and for the outcome (ischemic stroke), they are expressed as a logarithmic probability per 1 unit high. (**A**) Scatter plot of methylation site cg02631906 associated with ischemic stroke. (**B**) Forest plot of methylation site cg02631906 associated with ischemic stroke. (**C**) “Leave-one-out” analysis plot of methylation site cg02631906 associated with ischemic stroke. (**D**) Funnel plot of the association of methylation site cg02631906 with ischemic stroke. (**E**) Scatter plot of methylation site cg08483560 associated with ischemic stroke. (**F**) Forest plot of methylation site cg08483560 associated with ischemic stroke. (**G**) “Leave-one-out” analysis plot of methylation site cg08483560 associated with ischemic stroke. (**H**) Funnel plot of the association of methylation site cg08483560 with ischemic stroke. IS: ischemic stroke; MR: Mendelian randomization; SNP: single nucleotide polymorphism; SD: standard deviation.

## Data Availability

Datasets analyzed during the current study are available from the corresponding author upon reasonable request.

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
