# Peer review of "Ischemic Postconditioning Regulates New Cell Death Mechanisms in Stroke: Disulfidptosis"

_biomolecules, 2024, doi:10.3390/biom14111390_

Round 1
Reviewer 1 Report
Comments and Suggestions for Authors
This study investigated ischemic postconditioning (IPostC) as a potential strategy to reduce stroke-induced brain injury, with a focus on its interaction with disulfidptosis, a newly identified cell death pathway characterized by protein disulfide accumulation. We identified and expanded the list of disulfidptosis-related genes (DRGs) relevant to stroke, revealing key genes and their interactions. Using young adult male C57BL/6 mice as a stroke model and bioinformatics analyses (PCA, UMAP, and differential gene expression), we identified PRDX1 as a crucial gene. GSEA highlighted PRDX1's role in protective pathways against ischemic damage. We also explored PRDX1's regulatory mechanisms through a ceRNA network and drug sensitivity analysis, proposing new therapeutic approaches. Additionally, immune infiltration analysis linked PRDX1 to key immune cells, emphasizing its dual role in stroke progression and recovery. The role of PRDX1 in ischemic stroke is not new. In addition, stroke is a disease of the ageing people with their comorbidities, including obesity and animal model shall reflect this feature (see, doi: 10.1111/acel.12678). Unfortunately, the authors used young animals, and therefore, some of the results may not translate to the clinical situation. All these points shall be added to the manuscript as study limitations.
Comments on the Quality of English Language
Good for publication purposes
Author Response
Reviewer #1
Thank you so much for your positive and constructive comments, which have helped us to significantly improve the manuscript.
Comments#1:The role of PRDX1 in ischemic stroke is not new.
Response1: We agree with this comment that the role of PRDX1 is not new in stroke. However, we propose for the first time a new concept, that is, PRXD1 as a key gene for disulfidptosis and the mechanism of stroke. In addition, the protective mechanisms of IPostC remain largely unknown. This article confirms the high expression of IPostC remission PRXD1 in stroke, which is a new insight with clinical therapeutic significance in this paper.
Therefore, we have revised the paper accordingly and added the following paragraphs in the discussion section: “Although the role of PRDX1 in ischemic stroke has been mentioned in other studies, our research provides fresh insights by identifying key genes and networks related to stroke, especially within the context of IpostC.”
Comments #2: In addition, stroke is a disease of the ageing people with their comorbidities, including obesity and animal model shall reflect this feature (see, doi: 10.1111/acel.12678). Unfortunately, the authors used young animals, and therefore, some of the results may not translate to the clinical situation. All these points shall be added to the manuscript as study limitations.
Response2: We agree and have added a new paragraph to discuss the limitation in the revised manuscript:
One of the limitations of our study is the use of young adult mice as the animal model for investigating ischemic stroke and the effects of ischemic postconditioning (IPostC). Stroke predominantly affects older individuals who often present with various comorbidities, including obesity, hypertension, and diabetes, which can significantly influence the pathophysiology and outcomes of the disease. The young adult mice used in our experiments do not fully replicate these age-related factors, which may limit the translational relevance of our findings to the clinical setting. Specifically, the absence of age-related changes and comorbid conditions in our animal model may lead to differences in immune response, cellular aging, and recovery mechanisms compared to older, comorbid human patients. Consequently, while our study provides valuable insights into the molecular and cellular mechanisms of IPostC, further research using aged animal models that better reflect the clinical population is necessary to fully understand the potential therapeutic benefits of IPostC in stroke patients.

Reviewer 2 Report
Comments and Suggestions for Authors
The manuscript focuses on the exploration of the molecular mechanisms underlying a newly characterized form of cell death, first described in 2023. The research was conducted using advanced molecular biology techniques and bioinformatics analysis. However, the authors' claim that “no experimental animal study was ever conducted on studying DRG in stroke” appears to be inaccurate. Even a cursory review reveals existing publications on this topic, such as those by Rongxing Qin et al., 2023 (doi: 10.3934/mbe.2023838), and research from the same institution as the authors, including the work of Shan-Peng Liu et al., 2024 (doi: 10.2174/0115672026286243240105115419). Moreover, the definition of Disulfidptosis provided in the manuscript – “This process is characterized by the rapid depletion of nicotinamide adenine dinucleotide phosphate (NADPH) in cells and high expression of solute carrier family 7 member 11 (SLC7A11) during glucose starvation…” – suggests that stroke, which is associated with impaired glucose supply to cells, is a primary cause of Disulfidptosis activation.
The section “Materials and methods” lacks important details, such as the number of animals used in each experimental group. Additionally, the rationale behind the choice of this particular reperfusion model as a control for the selected occluding model is unclear. It would seem more appropriate to use a model involving the removal of the ligature after an extended period. In the chosen reperfusion model, the selected animal line demonstrates adaptive mechanisms to mild hypoxic damage, likely due to the vascular structure in these mice, which ensures adequate brain perfusion even under conditions of prolonged occlusion of the middle cerebral artery.
Given that the study relies on gene expression analysis, the timing of brain tissue sampling following occlusion or reperfusion is crucial, as is the specific brain region examined, presumably the cerebral cortex.
The statistical significance of the selected confidence threshold of >0.4 is questionable. The manuscript indicates that a significance level of less than 0.05 was used for KEGG gene set enrichment analysis with an FDR of <0.25. However, this threshold for FDR is generally not considered stringent enough to indicate the significance of the selected pathways. Typically, an FDR is deemed significant at a much lower threshold, such as <0.05.
On what basis were the specific immunological markers selected during the analysis of immune cell infiltration, and why were other potential markers excluded from consideration?
The inclusion of Figure 3 might be unnecessary.
Figure 4 presents 27 (16+11) DRG, yet the text refers to 18 previously identified DRG and 40 identified in total, including those found in this study. Clarification is needed regarding the identity of these 27 genes.
Figure 5 indicates that five DRG genes are crucial in distinguishing between SHAM, MCAo, and IPostC conditions. Although the results for these five genes meet the criteria for statistical significance, the decision to focus solely on PRDX1 for further analysis needs further justification.
Figure 6 illustrates that PRDX1 expression does not increase with IPostC, possibly due to the choice of the IPostC model, as mentioned earlier. Despite this, a “GSEA analysis of PRDX1 in stroke and IPostC” was conducted, the purpose of which is unclear, as is the rationale behind subsequent analyses. The manuscript would benefit from a more detailed explanation of why PRDX1 was chosen and why it is considered diagnostically valuable in conditions where its expression remains unchanged in IPostC.
Author Response
Reviewer #2
Thank you so much for your favorable comments and constructive feedback, which further enhanced our manuscript quality.
Comments #1:The reviewer points out that there are existing studies on this topic of disulfidptosis, including recent publications by Rongxing Qin et al. (2023) and research from the authors' own institution by Shan-Peng Liu et al. (2024).
Response1: We agree with your comment, and have revised the manuscript by citing previous literatures, including our own.
Comments #2:The section “Materials and methods” lacks important details, such as the number of animals used in each experimental group.
Response 2: We have added the animal number for each group in the revised manuscript.
Comments #3: Additionally, the rationale behind the choice of this particular reperfusion model as a control for the selected occluding model is unclear. It would seem more appropriate to use a model involving the removal of the ligature after an extended period. In the chosen reperfusion model, the selected animal line demonstrates adaptive mechanisms to mild hypoxic damage, likely due to the vascular structure in these mice, which ensures adequate brain perfusion even under conditions of prolonged occlusion of the middle cerebral artery.
Response 3: Thank you for your valuable feedback regarding the choice of reperfusion model in our study. We acknowledge the importance of selecting an appropriate control model to accurately reflect the clinical scenario. Our laboratory has extensive experience and a well-established protocol in constructing ischemic postconditioning (lPostC) models, as evidenced by our previous work (Xie et al., 2013; Joo et al., 2013). These models have been optimized over years of research to reliably reproduce the effects of ischemic postconditioning and facilitate the exploration of its underlying mechanisms. We have cited our previous studies in our revised manuscript.
We understand the concern regarding the potential adaptive mechanisms in the selected animal line that might ensure adequate brain perfusion even under prolonged occlusion. However, our current model has been chosen based on its reproducibility and the extensive data we have accumulated over time, which consistently demonstrates its relevance in studying ischemic postconditioning.
Comments #4:The statistical significance of the selected confidence threshold of >0.4 is questionable. The manuscript indicates that a significance level of less than 0.05 was used for KEGG gene set enrichment analysis with an FDR of <0.25. However, this threshold for FDR is generally not considered stringent enough to indicate the significance of the selected pathways. Typically, an FDR is deemed significant at a much lower threshold, such as <0.05.
Response 4: Thank you for your feedback regarding the statistical thresholds used in our analysis. We recognize the importance of selecting appropriate thresholds to ensure the robustness and reliability of our findings.
Regarding the protein-protein interaction (PPI) analysis conducted via the STRING database, we chose a medium confidence threshold of 0.4. This decision was made to maximize the inclusion of disulfidptosis-related genes (DRGs) in subsequent analyses, including pathway enrichment. While a higher confidence threshold, such as 0.7, could have been used, we prioritized a broader analysis to capture more potential interactions that might contribute to understanding the complex mechanisms involved. Importantly, the associations among DRGs were validated at the genetic level (see Figure 2A), and further supported by machine learning analyses that identified the most central genes, thereby justifying the use of a 0.4 threshold.
As for the gene set enrichment analysis (GSEA), we acknowledge that a false discovery rate (FDR) threshold of <0.05 is typically considered more stringent. However, we opted for an FDR threshold of <0.25, which is still commonly accepted in exploratory bioinformatics analyses, particularly when aiming to identify broader patterns and pathways that might otherwise be missed with stricter criteria. By using this threshold, we ensured that relevant pathways with potential biological significance were not prematurely excluded from our analysis. This approach allows for a more comprehensive examination of the data, capturing pathways that, while having a higher FDR, may still provide valuable insights into disulfidptosis-related processes.
We believe that our chosen thresholds strike a balance between sensitivity and specificity, allowing us to explore the data in depth while retaining the statistical rigor necessary for meaningful interpretation.
Comments #5: On what basis were the specific immunological markers selected during the analysis of immune cell infiltration, and why were other potential markers excluded from consideration?
Response 5: The immune cell infiltration analysis was designed to clarify the relationship between key genes, stroke, and the post-stroke immune adaptation mechanisms. Through the use of machine learning methods, PRDX1 emerged as the primary key gene of interest (Figure 4B-D), which guided our decision to focus on the immune infiltration analysis specifically for PRDX1. This focus allowed us to delve deeply into the role of PRDX1 in modulating the immune response during stroke.
We acknowledge Reviewer 2’s valuable suggestion to explore immune infiltration in relation to other potential therapeutic targets, such as ACTB, CYFIP1, APBB1IP, and PGD. In response, we have conducted additional immune infiltration analyses for these genes and have included the results in the manuscript (Figure S2). As noted in the revised RESULTS section, "Consistent results exist for other key genes, including ACTB, APBB1IP, CYFIP1, and PGD (Figure S2)." This expanded analysis further enriches our understanding of the immune response in stroke and supports the potential therapeutic relevance of these additional genes.
Comments #6:The inclusion of Figure 3 might be unnecessary.
Response 6: We agree and have decided to delete it in the main text and used it as a supplementary data (Figure S1).
Comments #7:Figure 4 presents 27 (16+11) DRG, yet the text refers to 18 previously identified DRG and 40 identified in total, including those found in this study. Clarification is needed regarding the identity of these 27 genes.
Response 7: Thank you for pointing out the need for clarification regarding the disulfidptosis-related genes (DRGs) presented in Figure 4. To address this, we have now included detailed data on the 27 DRGs associated with disulfidptosis in Table S3. This table provides comprehensive information on the identity and characteristics of these genes, helping to clarify their role and significance in our study
Comments #8:Figure 5 indicates that five DRG genes are crucial in distinguishing between SHAM, MCAo, and IPostC conditions. Although the results for these five genes meet the criteria for statistical significance, the decision to focus solely on PRDX1 for further analysis needs further justification.
Response 8: As mentioned in our response to Point 5, the focus on PRDX1 in our analysis was determined by the use of machine learning algorithms, specifically LASSO and Random Forest, which identified PRDX1 as the most significant gene among those analyzed. However, we fully acknowledge the importance of other key genes, such as ACTB, CYFIP1, APBB1IP, and PGD, which also showed significant roles in distinguishing between SHAM, MCAo, and IPostC conditions.
To address this, we have expanded the Discussion section to include a detailed explanation of the diagnostic and therapeutic potential of these additional genes. The text now reads:
“Study results suggest that ACTB, CYFIP1, APBB1IP and PGD also possess diagnostic and target value. ACTB, a key cytoskeletal component, is crucial in stroke pathology as its disruption impairs neuronal structure and function, affecting stroke outcomes and recovery [61, 62]. CYFIP1 plays a crucial role in stroke recovery by enhancing synaptic plasticity and dendritic remodeling through its regulation of CAMK2A expression, with disruptions in CYFIP1 impairing these neuroprotective effects [63]. APBB1IP (RIAM) is crucial for the proper activation and function of integrins in neutrophils, with its absence leading to impaired phagocytosis, with its dysregulation potentially impacting stroke-induced neuronal injury and repair [64]. PGD, an enzyme in the pentose phosphate pathway, is essential for managing oxidative stress [65], altered PGD expression can influence stroke outcomes by affecting the cell's oxidative stress response. Collectively, these genes are critical for understanding the mechanisms by which IPostC regulates stroke and for identifying potential therapeutic targets.”
Comments #9:Figure 6 illustrates that PRDX1 expression does not increase with IPostC, possibly due to the choice of the IPostC model, as mentioned earlier. Despite this, a “GSEA analysis of PRDX1 in stroke and IPostC” was conducted, the purpose of which is unclear, as is the rationale behind subsequent analyses. The manuscript would benefit from a more detailed explanation of why PRDX1 was chosen and why it is considered diagnostically valuable in conditions where its expression remains unchanged in IPostC.
Response 9: We appreciate the reviewer's concern regarding the expression of PRDX1 in the IPostC model and the rationale for conducting GSEA analysis despite PRDX1 expression not significantly increasing with IPostC. The decision to focus on PRDX1 was based on its established role as a key gene in response to oxidative stress and cellular damage, which are central to the pathophysiology of stroke. The rationale for conducting GSEA analysis on PRDX1 in this context was to further explore the molecular pathways and biological processes associated with PRDX1 expression in different experimental conditions. We have included the following additional content in the discussion section:
We recognize that PRDX1 expression did not increase in the IPostC group compared to the MCAo group, likely due to the specific IPostC model used. Despite this, PRDX1 was chosen for further analysis due to its critical role in the oxidative stress response, a key factor in stroke pathology. Although PRDX1 levels were reduced in the IPostC group, they remained elevated compared to the SHAM group, indicating ongoing residual damage. The GSEA analysis was conducted to explore the pathways associated with PRDX1 under different conditions, providing insights into its continued relevance in the context of IPostC and its potential as a diagnostic and therapeutic target.

Round 2
Reviewer 1 Report
Comments and Suggestions for Authors
The manuscript has been considerably improved. The authors have added a critical sentence acknowledging the use of young mice instead of old mice, but they did not provide a reference.
Comments on the Quality of English LanguageEnglish is good enough
Author Response
Reviewer #1
Thank you very much for your constructive comments again, which helped us greatly improve the manuscript.
Comments#1:The authors have added a critical sentence acknowledging the use of young mice instead of old mice, but they did not provide a reference.
Response1: Thank you for your valuable feedback regarding our manuscript. We have addressed your concern by adding a critical sentence in the discussion section to acknowledge the use of young mice instead of old mice, along with the corresponding reference [66] at line 616. We appreciate your insightful comments, which have helped us improve the clarity of our work.
We agree and add a new reference to complete the discussion of the article's limitations:
One of the limitations of our study is the use of young adult mice as the animal model for investigating ischemic stroke and the effects of IPostC. Stroke predomi-nantly affects older individuals who often present with various comorbidities, in-cluding obesity, hypertension, and diabetes, which can significantly influence the pathophysiology and outcomes of the disease [66].
Uzoni, A.; Popa-Wagner, A. Caloric restriction stabilizes body weight and accelerates behavioral recovery in aged rats after focal ischemia. Aging Cell 2017, 16, 1394-1403, doi:10.1111/acel.12678.
